# On Integrating Size and Shape Distributions into a Spatio-Temporal Information Entropy Framework

**DOI:** 10.3390/e21111112

**Published:** 2019-11-13

**Authors:** Didier G. Leibovici, Christophe Claramunt

**Affiliations:** 1School of mathematics and Statistics, University of Sheffield, Sheffield S3 7RH, UK; 2Naval Academy Research Institute, 29240 Brest CEDEX 9, France; christophe.claramunt@ecole-navale.fr

**Keywords:** spatio-temporal information, geolocated data, entropy decomposition, permutation entropy, patch size distribution, patch shape distribution, multiple scale, co-occurrences, spatio-temporal data analysis, multiway correspondence analysis, land cover change

## Abstract

Understanding the structuration of spatio-temporal information is a common endeavour to many disciplines and application domains, e.g., geography, ecology, urban planning, epidemiology. Revealing the processes involved, in relation to one or more phenomena, is often the first step before elaborating spatial functioning theories and specific planning actions, e.g., epidemiological modelling, urban planning. To do so, the spatio-temporal distributions of meaningful variables from a decision-making viewpoint, can be explored, analysed separately or jointly from an information viewpoint. Using metrics based on the measure of entropy has a long practice in these domains with the aim of quantification of how uniform the distributions are. However, the level of embedding of the spatio-temporal dimension in the metrics used is often minimal. This paper borrows from the landscape ecology concept of patch size distribution and the approach of permutation entropy used in biomedical signal processing to derive a spatio-temporal entropy analysis framework for categorical variables. The framework is based on a spatio-temporal structuration of the information allowing to use a decomposition of the Shannon entropy which can also embrace some existing spatial or temporal entropy indices to reinforce the spatio-temporal structuration. Multiway correspondence analysis is coupled to the decomposition entropy to propose further decomposition and entropy quantification of the spatio-temporal structuring information. The flexibility from these different choices, including geographic scales, allows for a range of domains to take into account domain specifics of the data; some of which are explored on a dataset linked to climate change and evolution of land cover types in Nordic areas.

## 1. Introduction

The Shannon entropy plays an important role as a descriptive statistic in various disciplines linked to the spatial domain, e.g., ecology, social sciences, urban planning [1,2,3,4] but often without entirely taking into account all the characteristics of the spatial or the spatio-temporal dimension as already proposed [5,6,7,8,9,10]. Nevertheless, the focus and motivation are often intended for the quantification the spatial or spatio-temporal structuring of the information provided by a categorical variable of interest. Entropy, as measuring the level of homogeneity and randomness, has been seen in the literature as a good candidate. There are many different alternative approaches to entropy, for example see [11,12] in the context of spatio-temporal clustering which can provide ways of understanding the structuring of the data, though, not necessarily with a direct way of quantifying it. Our purpose in this paper is to propose a framework that would reconcile classical approaches involving entropy as a metric with more recent literature [5,6,7,8,9,10]. The goal is how to better take into account the spatio-temporal embedding of the information that would accommodate an entropy approach.

In the classical approach, an underlying spatial ’structural support’ is usually considered, using a categorical variable *S* identifying a set of sub-regions of the whole studied region. For socio-economic studies *S* is often a fixed set of administrative units often linked to population sizes. In geographical studies it can be preferred to use either regular grids or elaborated units based for example on land conditions or climate, e.g., agro-ecological zoning systems, [13,14]. Then, for a given statistic that can be mapped to each sub-region (i.e., S=s), such as fraction of coverage of a specific land cover, frequency of unemployment, number of public buildings, it is possible to quantify the spatial structuration of that category *c* by the Shannon entropy:(1)$$H\left(c\left(S\right)\right)=-\sum_{s}p_{s|c}log\left(p_{s|c}\right)=H\left(S|C=c\right)$$
where $p_{s|c}=n_{sc}/n_{c}=p_{sc}/p_{c}$ is the proportion of *c*’s that are in the sub-region *s*, i.e., $n_{sc}$ is the number of entities with the characteristic *c* found in sub-region *s* among $n_{c}$ in the whole population of entities *N* (e.g., persons). For land cover $p_{s|c}$ is the fraction of land occupation in *s* but relative to the whole studied area; one might be also interested in the $p_{c|s}$, the fraction of *c* within *s* and so the entropy H(C∣S=s). Note the notation $p_{sc}$ is without ambiguity referring to the joint distribution of *S* and *C*, $p_{SC}=\left\{p_{sc};\left(s,c\right)\in S\times C\right\}$ (a matrix), as well as $p_{c}=\sum_{s}p_{sc}$ to the distribution of variable *C*, i.e., the vector $p_{C}=\left(\ldots ,p_{c},\ldots \right)$ and idem for $p_{s}$. So, Equation (Equation 1) is the entropy of the conditional distribution of the categorical variable *S*, knowing C=c. The categorical variable *C* expresses *c* as one of its category, e.g., *C* corresponds to a land cover classification, to a socio-professional indicator, to a building typology or a simple dichotomy between cases and at risk of a specific disease.

It provides a quantification describing the repartition of each single category *c*, spatially across the sub-regions, e.g., the entropy is maximum if the distribution is uniform, or, reaching very small values when segregation in a few sub-regions occurs (0 if *c* is concentrated in only one sub-region *s*). However, the spatial organisation of the region in *s* sub-regions is not taken into account. Any permutation of the values would give the same entropy, only the semantic attached to the sub-regions is rooting down a spatial understanding for *c*. Nonetheless, a sub-region system as such, often represents a level of aggregation of the observations within each sub-regions *s*. The number of sub-regions may be too small to convey sufficient statistical information about *topological* information between the sub-regions and multiple scale integration may be looked for in the regioning system. Here, ‘topological’ is understood as spatial organisation and configuration, e.g., proximity, connection, homogeneity, between and within observations or units where the observations are in. The interplay of proximities of categories and multiple co-occurrences have been proposed to define spatial entropy and spatio-temporal entropy measures [15] but do not pertain easily spatial or spatio-temporal graphical representations even though local indices are possible. From (Equation 1) a further decomposition of the bivariate information S,C (see Section 2) expresses the role of the spatial support *S*. Despite not being able to fully provide a spatial entropy measure for *C*, it is a useful tool when focusing on characterising a regional system *S* or comparing two regional systems *S* and $S^{\prime }$, say encompassing a change of scale, for a range of economical and socio-demographic variables (i.e., a series of variables like *C*). Questions such as “which spatial scale provides the most or least disparity?” can then be approached. It is not particularly useful when no *a priori* spatial regional system makes sense as in landscape ecology but the scale aspect does. So, the decomposition approach constitutes a basis for a framework to the spatial or spatio-temporal information related to *C* when using appropriate spatial or spatio-temporal descriptors that leads to a range of spatio-temporal entropy measures (see Section 3).

Landscape ecology has provided a range of spatial and topological descriptors, e.g., richness, adjacency, patchiness, connectivity, that help to describe how the spatio-temporally information from a categorical variable *C* is organised and its role into understanding associated ecological processes [1,16], including the role of entropy [17]. The temporal evolution of sizes and shapes of patches *per* categories of a variable *C* are the consequences of the underlying spatio-temporal processes involved. Therefore, depicting the information structuring of these spatial-temporal descriptors in interaction, using the entropy, would contribute to this endeavour. Instead of using external spatial descriptors, linked to a fixed spatial support as with the above description of *S*, this paper proposes to use the variables patch-size, Si and patch-shape Sh to be combined with the information from *C* in order to decompose their joint entropy.

A spatial patch can be defined as an homogeneous zone according to a category *c* and can be also understood as a cluster. When observations are recorded *per* elementary units with proportions falling in that unit (also known as compositional data), a patch may be defined using a minimum proportion for the same category *c*, i.e., enough observations with a category *c* in one unit then considered as a patch or part of a bigger patch. For compositional data, the patch can take into account a fuzziness (as a degree of membership of a patch) due to decreasing values of the proportion of the category *c*. Note that with such compositional data, patches of different categories may then overlap. Depending on the modelling choice, separation of the patches can be operated, for example using the dominant category among the categories in the overlapping patches.

Similarly to the spatial structuration, Equation (Equation 1) can also be written for *T* a time structure of the observations, with *t* being a sub-period of the whole time period of observations defined by the categorical variable *T*. Order and proximities of the *t*s allow to define a patch as homogeneous temporal zone according to a category *c* from *C*. A temporal patch is then also associated with variable descriptors such as Ti for a temporal patch size and temporal patch shape. With compositional data, a temporal ’shape’, Th can take the form of a pattern of increasing and decreasing proportion values which becomes close to the notion of motifs, i.e., succession of specific categories. The latter can be also achieved from borrowing concepts involved in permutation entropy [18,19,20] to integrate time flow in the dynamic of the categories, e.g., increase of a proportion of a category *c* from past to future, motif as increase followed by a decrease, motifs due to pre-defined possible successions of categories. Therefore, Size and shape of patches of *C* are seen here as the basics of the spatio-temporal structuration of *C* applicable in various domains, e.g., physical geography, social geography, demography etc. For a land cover data, knowing the different sizes and shapes for a particular vegetation configuration will help understanding its ecology, e.g., invasive species; in urban planning these sizes and shapes will contribute to analyse social segregation and in epidemiology, sizes and shapes may relate to contagion paths and outbreak mechanisms.

The paper proposes a framework approach integrating the Shannon decomposition theorem (Section 2) using these spatio-temporal descriptors. The *modus operandi* of this framework is detailed in Section 7 and illustrated with a land cover evolution data in Section 8. The three major steps are: ***(i)***
*defining patches rules*, ***(ii)** extracting the multiway information crossing spatio-temporal patch characteristics and C*, and, ***(iii)** quantifying and mapping the spatio-temporal information from entropy decomposition and related methods*. This framework, termed the *patch size and shape entropy* (PsishENT) framework, is based on the Shannon entropy and existing spatio-temporal approaches of the Shannon entropy itself [6,8,15] on the rendered information in (ii) (Section 3). As part of (iii) a multiway correspondence analysis can be used [21,22] (Section 5) which is related to the concept of mutual information reminded in Section 2. This multiway analysis provides a decomposition for which, each part has an interpretation similar to a product of the spatial, temporal and categorical distributions, therefore providing after a transformation a simple entropy decomposition (see Section 2 and Section 5). These three major steps of the framework are detailed within their potential sub-steps in the next few sections before summarising the approach in Section 7.

## 2. Using Shannon’s Multivariate Decomposition Entropy

Equation (Equation 1) shows that the historical approach into a spatial entropy introduced the conditional entropy as a natural way forward. When considering all the categories of the variable of interest *C*, so expending for all *c*’s of *C* and using the joint entropy, (Equation 1) becomes: (2)$$\begin{array}{ccc} H(C,S) & {=}^{def} & -\sum_{s,c}p_{sc}log\left(p_{sc}\right)\\ & = & -\sum_{c}p_{c}log\left(p_{c}\right)-\sum_{c}p_{c}\sum_{s}p_{s|c}log\left(p_{s|c}\right) \end{array}$$(3)$$\begin{array}{cc} = & H(C)+H(S|C)\\ = & -\sum_{s}p_{s}log\left(p_{s}\right)-\sum_{s}p_{s}\sum_{c}p_{c|s}log\left(p_{c|s}\right) \end{array}$$(4)$$\begin{array}{cc} = & H(S)+H(C|S) \end{array}$$
known as the entropy decomposition theorem [23,24], where the roles of *S* and *C* in the bivariate distribution can be swapped as expressed by Equations (Equation 3) and (Equation 4). Note that this presentation is not limited to the spatial context and *S* or *C* can be any categorical variables. H(C) is then the entropy for the overall distribution of the *c* categories of the variable *C* in the considered region, without explicit integration of the role of the spatial dimension. H(S∣C) is the mathematical expectation of Formula (Equation 1) over all *c* values and expresses the role of *C* in the potential structuration of the sub-regions, i.e., if H(S∣C) is small then *C* contributes substantially in highlighting differences (non uniformity) in *S*. It implies a spatial configuration due to *C* in the sub-regions but without knowing which categories are the most involved. The decomposition involving H(C∣S), expressing how *S* contributes in describing *C* distribution, or, how *S* influences *C* non-uniformity, might be more interesting in representing spatially the impact of the variable *C* for example by visualising the *S* sub-regions using the statistic $H_{C|s}^{ratio}{=}^{def}p_{s.}H\left(C|S=s\right)/H\left(C|S\right)$, for each sub-region *s*. This normalisation called from now on, *conditional entropy ratio*, is a normalisation adapted to the analysis of parts of the conditional entropy.

A normalisation of the Shannon entropy such as $H^{u}\left(S\right){=}^{def}-1/log\left(|S|\right)\sum_{s}p_{s}log\left(p_{s}\right)$, allows to get a span between 0 and 1, i.e., 1 for ’completely’ uniform (*u*) distribution. If the former normalisation (ratio) has the advantage of being self-referring, mapping $H^{u}\left(C|S=s\right)=1/log\left(|C|\right)H\left(C|S=s\right)$ is independent of the number of categories used and allows sub-regions comparisons and the above statistic is the same:(5)$$\begin{array}{ccc} H_{C|s_{1},s_{2},\ldots }^{ratio} & {=}^{def} & \left(p_{s_{1}.}H\left(C|S=s_{1}\right)+p_{s_{2}.}H\left(C|S=s_{2}\right)+...\right)/H\left(C|S\right)\\ & = & \left(p_{s_{1}.}H^{u}\left(C|S=s_{1}\right)+p_{s_{2}.}H^{u}\left(C|S=s_{2}\right)+...\right)/H^{u}\left(C|S\right)\\ & = & H_{C|s_{1},s_{2},...}^{u-ratio} \end{array}$$

Using the normalisation respective to uniform distribution, Equations (Equation 3) and (Equation 4) become: (6)$$\begin{array}{ccc} H^{u}\left(C,S\right) & = & -1/\left(log(|S|\right)+log(|C|))\sum_{s,c}p_{sc}log\left(p_{sc}\right) \end{array}$$(7)$$\begin{array}{cc} = & \frac{log(|C|)}{log(|S|)+log(|C|)}H^{u}\left(C\right)+\frac{log(|S|)}{log(|S|)+log(|C|)}H^{u}\left(S|C\right) \end{array}$$(8)$$\begin{array}{cc} = & \frac{log(|S|)}{log(|S|)+log(|C|)}H^{u}\left(S\right)+\frac{log(|C|)}{log(|S|)+log(|C|)}H^{u}\left(C|S\right) \end{array}$$

The decomposition theorem of the entropy is not specific to *S* and *C*, only a bivariate imformation is required. Recently [10] used the entropy decomposition theorem with a bivariate information referring to the categories, *C* and spatially adjacent categories by then allowing a decomposition of the entropy of the spatial contiguity of categories from the adjacency distribution, i.e., similarly to co-occurrences of order 2, [6].

Using Equation (Equation 3), one gets [H(C,S)−H(S∣C)]−H(C∣S)=H(C)−H(C∣S) which from Equation (Equation 4) is also [H(C,S)−H(C∣S)]−H(S∣C)=H(S)−H(S∣C) therefore:(9)$$H\left(C\right)-H\left(C|S\right)=H\left(S\right)-H\left(S|C\right){=}^{def}MI\left(C,S\right)$$
defining the Mutual Information (MI) between the two variables *c* and *S*. Then from Equation (Equation 3) or (Equation 4):(10)H(C,S)=H(S)+H(C)−MI(C,S)
leads to another way of defining the mutual information that is by the Kullback-Leibler divergence between $p_{SC}=\left\{p_{sc};\left(s,c\right)\in S\times C\right\}$ and $p_{S}⊗p_{C}=\left\{p_{s}p_{c};\left(s,c\right)\in S\times C\right\}$, i.e., the joint distribution and its approximation under the hypothesis of independence,

(11)$$D_{KL}\left(p_{SC}|p_{S}⊗p_{C}\right)=\sum_{sc}p_{sc}log\left(p_{sc}/\left(p_{s}p_{c}\right)\right){=}^{def}MI\left(C,S\right)$$

From Equations (10) and (11), if *S* and *C* are statistically independent, i.e., $p_{sc}=p_{s}p_{c}$, or similarly the *c* profiles in different sub-regions are all the ’same’ (proportionals), then we have additivity of their respective entropy when considering the joint information. It does not mean that *C* is not structured spatially, only that the structuration *S* is expressing a common spatial structure (irrespective to *c*’s). Another structuration $S^{\prime }$ might reflect otherwise.

### With Spatial and Temporal Supports

The entropy decomposition theorem, in the form of Equation (Equation 10), is easily extendable to multivariate situations, within a spatial or non-spatial context:(12)$$H\left(C_{1},C_{2},...,C_{p}\right)=\sum_{v=1}^{p}H\left(C_{v}\right)-MI\left(C_{1},C_{2},...,C_{p}\right)$$
for *p* categorical variables $C_{1},C_{2},\ldots ,C_{p}$, with the conceptually easily generalisable mutual information of the *p* variables: $MI\left(C_{1},C_{2},...,C_{p}\right)=D_{KL}\left(p_{C_{1}C_{2}...C_{p}}|p_{C_{1}}⊗p_{C_{2}}...⊗p_{C_{p}}\right)$. Within a spatio-temporal context for one categorical variable *C*, this takes the form: (13)$$\begin{array}{ccc} H(C,S,T) & = & H(C)+H(S)+H(T)-MI(C,S,T)\\ & = & H(S,T)+H(C|S,T)=H(C)+H(S,T|C) \end{array}$$(14)$$\begin{array}{cc} = & H(S)+H(T|S)+H(C|S,T)\\ = & H(S)+H((C,T)|S)\\ = & H(S)+H(T|S)-H(S|T)+H(C,S|T) \end{array}$$(15)$$\begin{array}{cc} = & MI(S,T)+H(T|S)+H(C,S|T) \end{array}$$(16)$$\begin{array}{cc} = & MI(S,T)+H(S|T)+H(C,T|S) \end{array}$$
generalising Equation (Equation 4) or (Equation 3).

These different formulations provide ways of decomposing and representing graphically each component as patterns, e.g., a map of the $H^{u}\left(C|S=s,T=t\right)=1/log\left(|C|\right)H\left(C|S=s,T=t\right)$ for all *s* at chosen *t* (intervals or sub-periods) or as time series plot at chosen sub-regions *s*.

## 3. Taking into Account Spatio-Temporal Relative Proximities

The structuration of the observations from knowing their distribution jointly for *S*, *T* and *C* leads to the multivariate decomposition theorem of the classical Shannon entropy but again no topological properties are really involved. However, as only the three-way data table S×T×C containing the distribution of occurrences of observations is used, it is also possible to use a distribution co-occurrences instead [6,15]. By then, the decomposition theorem will be framed within a spatio-temporal entropy measure. For a chosen order of co-occurrence *k*, counting the number of co-occurrences among the observations $o_{i}$ with $C(o_{i})=c$ is made from considering the observations in a manifold $E_{st}$ within an Euclidean space, e.g.,:(17)$$\begin{array}{c} o_{1},o_{2},o_{3}\in E_{st}\;\mathrm{are}\;\mathrm{in}\;\mathrm{co}-\mathrm{occurrence}\;\mathrm{of}\;\mathrm{order}\;k=3\;\mathrm{for}\;C=c,\\ \;\mathrm{if}\;max_{o,o^{\prime }\in \left\{o_{1},o_{2},o_{3}\right\}}d\left(o,o^{\prime }\right)\leq d_{\epsilon }\\ \;\mathrm{where}\;d\;\mathrm{being}\;\mathrm{the}\;\mathrm{distance}\;\mathrm{used}\;\mathrm{in}\;\mathrm{all}\;\mathrm{cells}\;stc\;\\ \mathrm{and}\;d_{\epsilon }\;a\;\mathrm{chosen}\;\mathrm{collocation}\;\mathrm{distance}\;\mathrm{parameter}. \end{array}$$

From this three-way table of counts of co-occurrences, a three-variate distribution of co-occurrences [6] is achieved, i.e., a spatio-temporal distribution of *C* that can be used with the Shannon entropy decompositions, i.e., Equations (Equation 13) to (Equation 16). For each cell stc of the three-way data table S×T×C, any non-negative indicator positively correlated, across st, with count of observations can also lead to a three-variate distribution-like table that can be used with the Shannon entropy decompositions formula, e.g., a local version of the distance-ratio weight used in [5]:$$d_{stc}^{ratio}{=}^{def}\frac{mean_{(o_{1},o_{2})\in W}d\left(o_{1},o_{2}\right)}{mean_{(o_{1},o_{2})\in B}d\left(o_{1},o_{2}\right)}$$

(18)$$\begin{array}{c} \;\mathrm{where}\;W=\{\left(o_{1},o_{2}\right)\in E_{st}\times E_{st}|C\left(o_{1}\right)=c,C\left(o_{2}\right)=c\}\\ \;\mathrm{and}\;B=\{\left(o_{1},o_{2}\right)\in E_{st}\times E_{st}|C\left(o_{1}\right)=c,C\left(o_{2}\right)\neq c\} \end{array}$$

The local computation within each $E_{st}$, of co-occurrences distributions, or of distance-ratio weights are subject to a border effect that is not encountered with the occurrences distributions. However, it is easy to modify formulations (Equation 17) or (Equation 18) to allow overlaps but enforcing at least one of the $o_{i}$ to be in $E_{st}$ and the others within a small distance, $d_{b}$, to the border. That distance needs to be smaller than $d_{\epsilon }$, by then minimising the over-count of co-occurrences, and, if $d_{b}$ is relatively smaller than the average distance between two observations in st, the estimation of $d_{stc}^{ratio}$ will not be too affected, i.e., proximities across the border will be taken into account without smoothing too much the values across neighbouring $E_{st}$’s. Without these overlaps, there could be under-estimation for the co-occurrences or distance-ratio statistics when a large number of observations are made close to borders.

### With a Symmetric or Non-Symmetric Spatio-Temporal Approach

In integrating the spatio-temporal approach of co-occurrences, the approach taken in the previous sub-section has been non-symmetric. Multiple observations were identified first with their category *c*, then their geolocation, spatio-temporally were taken into account within a $E_{st}$, i.e., a semantic bias was focusing on the *c*’s observations scattered spatio-temporally. So, in definitions (Equation 17) or (Equation 18) the distances were spatial distances at time *t* within the sub-region *s*, $E_{st}$. To be fully symmetric the co-occurrence definition needs to be:(19)$$\begin{array}{c} o_{1},o_{2},o_{3}\in E_{stc}=\left\{o\in S\times T\times C|\;C\left(o\right)=c,S\left(o\right)=s,T\left(o\right)=t\right\}\\ \;\mathrm{are}\;\mathrm{in}\;\mathrm{co}-\mathrm{occurrence}\;\mathrm{of}\;\mathrm{order}\;k=3,\\ \;\mathrm{if}\;max_{o,o^{\prime }\in \left\{o_{1},o_{2},o_{3}\right\}}d\left(o,o^{\prime }\right)\leq d_{\epsilon }\\ \;\mathrm{where}\;d()\;\mathrm{being}\;\mathrm{the}\;\mathrm{distance}\;\mathrm{in}\;S\times T\times C\;\\ \mathrm{and}\;d_{\epsilon }\;a\;\mathrm{chosen}\;\mathrm{collocation}\;\mathrm{distance}\;\mathrm{parameter} \end{array}$$

In the definition (Equation 19), S is the spatial dimensional space in which the regional system *S* is embedded, similarly for T as a temporal dimensional space and C a variable space where categorical variables can be expressed. The distances in S and T are the natural Euclidean distances and in C, proximities can be expressed as 0 or 1 or using a dissimilarity taking into account closeness between categories. Then, a distance in S×T×C has to be chosen, e.g., sum of the distances in each dimension, their product, their maximum?

The equivalence of this definition to the former definition in (Equation 17) for particular settings highlights in fact the substantial conceptual difference. Implicitly, in definition (Equation 17) there was no distance *per se* for time *T*, $E_{st}$ being a snapshot of the spatial sub-region *s* at time *t*, neither for categories *C*, i.e., implicit infinite distance for different categories or times, making the two definitions equivalent. Combining arithmetically distances in each sub-space or building a multidimensional distance is not straightforward due to the different scales and semantics involved. Therefore, it might be more appropriate to use a distance-rule across the three spaces S, T, C, such as:(20)$$d\left(o,o^{\prime }\right)\leq d_{\epsilon }\Leftrightarrow \left\{\begin{array}{c} d_{S}\left(o,o^{\prime }\right)\leq d_{\epsilon_{S}}\\ d_{T}\left(o,o^{\prime }\right)\leq d_{\epsilon_{T}}\\ d_{C}\left(o,o^{\prime }\right)\leq d_{\epsilon_{C}} \end{array}\right.$$
instead of a distance in S×T×C. Noticeably, the definition (Equation 19) establishes now a co-occurrence not just for *c* but *s* and *t* too, as a joint category (s,t,c), then from (Equation 20), the criterion $max_{o,o^{\prime }\in \left\{o_{1},o_{2},o_{3}\right\}}\left[d\left(o,o^{\prime }\right)\right]\leq d_{\epsilon }$ is enough to record a co-occurrence of observations, here of order k=3. However, the co-occurrence "of what?" can take different forms. The first line in definition (Equation 19) is modulated with set of chosen rules, i.e., the set of strict values in (Equation 19) are complemented by another distance-rule based criterion, allowing to adopt multiple categorisations of the co-occurrence, therefore multiple co-occurrences at once. For example, if for each pairs of observations in the co-occurrence (of order k=3), $d_{S}\left(o,o^{\prime }\right)\leq d_{r_{S}}<d_{\epsilon_{S}}$, then $S(o_{1})$, $S(o_{2})$ and $S(o_{3})$ are valid spatial categorisations (*S*) for this co-occurrence, idem with *T* and *C*. This sort of fuzzy characterisation effectively removes the problem of the ’border effect’ mentioned in the previous section. The majority across each categorical variable could also characterise a co-occurrence, e.g., $o_{1},o_{2},o_{3}$ satisfying definition (Equation 20) and $o_{1}$ with $\left(s,t^{\prime },c\right)$, $o_{2}$ with $\left(s^{\prime },t,c\right)$, $o_{3}$ with $\left(s^{\prime },t^{\prime },c^{\prime }\right)$ giving a categorisation of the co-occurrence as $\left(s^{\prime },t^{\prime },c\right)$, so not necessarily reflecting any of these observations.

Similarly, the local distance-ratio weight definition is asymmetric by essence but *S* or *T* can be focused on, not just *C*. A fully symmetric version, looking at categories defined as stc, leads to indicators that can take various forms depending on the choice of distances, e.g., closer to its definition as global indice [5], or to its spatio-temporal version [25,26]:$$d_{stc}^{ratio}{=}^{def}\frac{mean_{(o_{1},o_{2})\in W}d\left(o_{1},o_{2}\right)}{mean_{(o_{1},o_{2})\in B}d\left(o_{1},o_{2}\right)}$$

(21)$$\begin{array}{c} \;\mathrm{where}\;W=\{\left(o_{1},o_{2}\right),o_{1},o_{2}\in S\times T\times C|\exists \;o\in O_{stc},\;d\left(o_{1},o\right)\leq d_{W}\;\mathrm{and}\;d\left(o_{2},o\right)\leq d_{w}\}\\ \;\mathrm{and}\;B=\{\left(o_{1},o_{2}\right),o_{1}o_{2}\in S\times T\times C|\exists \;o\in O_{stc},\;d\left(o_{1},o\right)\leq d_{B}\;\mathrm{and}\;\forall \;o\in O_{stc},d\left(o_{2},o\right)\geq d_{B}\}\\ \mathrm{given}\;O_{stc}=\left\{o\in S\times T\times C|C\left(o\right)=c,S\left(o\right)=s,T\left(o\right)=t\right\} \end{array}$$

From playing symmetrical roles in the data table S×T×C, as it does for the occurrence distribution used for the joint Shannon entropy, Equations (Equation 13) to (Equation 16) can be fully expressed within the spatio-temporal entropy approaches of *k*-co-occurrences or localised indices such as the distance-ratio. As a consequence when replacing *S* and *T*, the structural framework of sub-regions and calendar chunks, by topological descriptors of *C* such as patches size or shapes, allows the framework to study directly spatio-temporal topological interactions of *C*, i.e., topological relations between a labelling from *C* with a spatial labelling from *C* and a temporal labelling from *C*.

## 4. Constructing the Spatial and Temporal Patches Characteristics

Considering of spatial and temporal patches as embedding the spatio-temporal structuring context for Section 2 has a twofold outcome. First, from categorising spatio-temporally the variable of interest *C*, it enables to relate different parts of the entropy decomposition to the spatial or the temporal or the spatio-temporal processes involved with *C*. Second, it allows a topological interpretation compatible with the spatio-temporal entropy approaches with proximities from Section 3.

The data structure concerning the spatio-temporal distribution for the categorical variable *C* is either a compositional data *per* areal units or a set of single observations, each available at a point or areal unit. For a compositional data, a vector of the counts for each category represents the distribution of *C* in each unit. In the case of single observations only a single value from C=c is an attribute of that observation. In the following of the paper, these will be termed *compositional* data and *observational* data respectively; without further description an observation will refer to both types.

The spatial or temporal patch criteria once established, patch size and patch shape can be defined accordingly. The categorical variables $S_{P}$ and $T_{P}$ will *identify* spatial and temporal patches across all *c*’s. As defined in the introduction, the generic definition of a patch is about connected observations of the same category. For compositional data, a chain or group of adjacent units will make a patch with a minimum proportion of *c* in each unit. For observational data, the connection of the observations with *c* have to be established using distance threshold (spatially, temporally or spatio-temporally). Then a patch is the set of points (or basic geometries) that encapsulate the observations which can be identified as the graph of the connected observations or by the convex hull of the observations or any other shape containing these observations. For both types of data, overlaps of patches may occurs. The patch size is defined by the count observations being part of, or falling into, the patch. Those remarks are valid for spatial and temporal patches $S_{P}$ and $T_{P}$ and define Si and Ti as patch size categorical variables. Note that if the range of sizes values is too large, groups of sizes may be defining the categories in Si and Ti.

With this generic definition of patches, shapes will be referring to the 2D geometry of the patch for spatial aspects and 1D geometry for time. When fuzziness of the patch is taken into account, for example with a proportion above a minimum required to be qualified as patch of *c*’s for compositional data or with a semantic distance across *c* categories for observational data, 2D+1 geometry and 1D+1 geometry are describing the shape. The +1 reflects the degree of membership. They can be referred as flat patterns (2D or 1D) or profile patterns (2D+1 and 1D+1). If for 1D no specific shape categorisation can be made, with 2D and 2D+1, clustering the shapes from geometric measures such as perimeter, volume, principal axes compactness, etc. can be used to further categorise the shape to be used as Sh.

Motifs, defined when the patch criterion includes the possibility of having more than one category *c* in the patch, from proximity relations, define other types of shapes. A spatial motif may be for example, the shape of a patch with two categories, $c_{1}$ and $c_{2}$ with $c_{1}$ being dominant (related to size), the motif with $c_{2}$ dominant being more likely to be included as well. It can also involves a topological relation, e.g., $c_{1}$ most often in the North of $c_{2}$, or $c_{1}$’s surrounded by $c_{2}$’s. It can corresponds to a patch composite as suggests the latter examples. A temporal motif may be a sequence of first $c_{1}$ observations for a number of time units followed by a number of time units with $c_{2}$, etc. The definition of the categories of shapes, as pattern, as motifs or both is of course a matter of the application in ecology, in economy, or epidemiology, as well as the level of complexity desired [20,27].

Focusing on the temporal dimension, the permutation entropy can be modulated by a distance, a meaningful difference, between observations when assessing their order, and so the occurrences of specific permuted patterns. This fuzzy assessment of the order is important when willing to separate really meaningful changes from smaller random changes. A similar refinement of the patterns or motifs has been proposed in [20] with an example on distance to the mid point within a pattern of length 3. For a given time series, the members of a permutation class $\pi_{k}$ can be defined as:(22)$${\left(x_{t},\;t=0,...,\left(N-1\right);\tau ;l=3\right)}_{\pi_{k}}=\left\{\left(x_{t+\tau },x_{t+2\tau },x_{t+3\tau }\right)|x_{t+\pi_{k}\left(1\right)\tau }\leq x_{t+\pi_{k}\left(2\right)\tau }\leq x_{t+\pi_{k}\left(3\right)\tau }\right\}$$
where $\pi_{k}$ refers to one of the 3! permutations of the triple (1,2,3), implicitly referring to the length of the pattern *l* with a lag τ. For example, if three values are ranked like $x_{t+\tau }\leq x_{t+3\tau }\leq x_{t+2\tau }$ then the triplet belongs to the pattern or motif of the permutation $\pi_{k}\left(1,2,3\right)=\left(1,3,2\right)$. It is a sequence with an increase between (t+τ) and (t+2τ) and a decrease between (t+2τ) and (t+3τ) to a value higher than (t+τ). In [20], two groups for any permutation are differentiated, if $d\left(x_{t+\pi_{k}\left(3\right)\tau },x_{t+\pi_{k}\left(2\right)\tau }\right)\geq d\left(x_{t+\pi_{k}\left(2\right)\tau },x_{t+\pi_{k}\left(1\right)\tau }\right)$ or not, making $\pi_{k}=123$ into a 123t representing a larger increase followed by a smaller (relatively to mid point) and 123b representing a smaller increase followed by a larger. For categorical variables, this presentation supposes either there is a predefined ordinal relationship between the categories or a compositional data where the motifs are worked on the proportions of a given category *c*. The permutation approach ensuring that all alternatives motifs are to be used in the entropy is not necessary or always welcome. Rules to define a range of specific patterns can replace the full permutation approach. Besides varying the parameter τ and *l* one may be interested in simple patterns of increase or decrease with l=2 but also allowing the patches to join up for various length of increases or decreases, i.e., when Ti becomes prominent.

The categorical variables Si, Sh, Ti and Th are replacing spatial and temporal categorisation of *S* and *T*. They are not used any more to pinpoint an observation in the time flow of space but characterise spatio-temporally the ’locality’ of where and when the observation occurred. The goal of this ’locality’ will be to encompass the local ’topology’ in space and time that is induced by the observations of *C* in the neighbourhood. Then, the spatio-temporal support exogenous to *C* processes disappears to become an inherent part of *C*. Note that the categorical variables $S_{P}$ and $T_{P}$ can also be considered as background information, a the spatio-temporal ’support’ similar to what *S* and *T* were providing but with the fundamental difference that $S_{P}$ is changing across time and $T_{P}$ across the space. Therefore they can be used directly only for entropy decomposition only at a specific time for $S_{P}$ or specific spatial unit for $T_{p}$ but also within a ’cumulative’ approach, e.g., $S_{P}$ describing all the set of all spatial patches at given times.

Once a set of specific topological characteristics linked to the spatio-temporal distribution of *C* are chosen, the joint distribution is established, from occurrences along with various choices of ’counting’ statistics leading to the three-variate distribution of interest (Section 3) and the entropy decomposition theorem(s) (Section 2) can be used. The next section proposes an alternative decomposition setting on which entropy can apply.

## 5. Using Multiway Correspondence Analysis

An important part of the PsishENT framework comes from the fact that the Shannon decomposition theorem(s) of Section 2 is based on working out a joint distribution to produce from the observations the multiway contingency table before using conditional probability properties. Equations (Equation 12) and (Equation 13), involving the mutual information, reflect the role played by the statistical independence of the categorical variables involved to build the joint distribution. Therefore, analysing the structure of independence of the multiway contingency table representing this joint distribution contributes to the spatio-temporal characterisation induced by *C*. The correspondence analysis of a two-way contingency table [28,29] provides a decomposition of the $\chi^{2}$ statistic of independence using a Singular Value Decomposition (SVD) of a specific matrix: $\sum_{r}\sigma_{r}^{2}=1+\chi^{2}/N$, where the $\sigma_{r}$s are the singular values of the matrix of the $p_{ij}/\left(p_{i}p_{j}\right)$, using the vectors $p_{I}$ and $p_{J}$ as weights in the sum of squares and inner product for each dimensional variables *I* and *J* [21]. In [21], this presentation has been extended to analysing a multiway table using tensor algebra as an extension of matrix calculus. The decomposition, say for a generic three-way data contingency table, of the $p_{stc}/\left(p_{s}p_{t}p_{c}\right)$ for (s,t,c)∈S×T×C (where *S*, *T* and *C* are here taken as generic categorical variables in the PsishENT framework, e.g., Si, Th, *C*), and $p_{stc}$ being a normalised measure correlated to the proportion of occurrences for the observations with categories *s*, *t* and *c* can be written:(23)$$p_{STC}/\left(p_{S}⊗p_{T}⊗p_{C}\right)=1+\sum_{r}\sigma_{r}\left(v_{S_{r}}⊗v_{T_{r}}⊗v_{C_{r}}\right)$$
where $\forall r,\;∥v_{S_{r}}{∥}_{p_{S}}^{2}=\sum_{s}p_{s}v_{{S_{r}}_{s}}^{2}=1$ and similarly for the other component vectors. Equation (Equation 23) can be written:(24)$$\begin{array}{ccc} p_{STC} & = & \left(p_{S}⊗p_{T}⊗p_{C}\right)+\sum_{r}\sigma_{r}\left(p_{S}v_{S_{r}}\right)⊗\left(p_{T}v_{T_{r}}\right)⊗\left(P_{C}v_{C_{r}}\right)\\ & = & \sum_{r=0}\sigma_{r}\left(p_{S}v_{S_{r}}\right)⊗\left(p_{T}v_{T_{r}}\right)⊗\left(p_{C}v_{C_{r}}\right) \end{array}$$
where $v_{S_{0}}$, $v_{T_{0}}$, and $v_{C_{o}}$ are the vectors of 1’s with corresponding dimensions, e.g., $1_{S}=\left(1,\ldots ,1\right)$ of length the number of categories in *S*, and $\sigma_{0}=1$. As in the SVD, the $\sigma_{r}^{2}$ are the maximum weighted sum of squares of a projection of the tensor $p_{STC}/\left(p_{S}⊗p_{T}⊗p_{C}\right)$ onto rank-one tensors (u⊗v⊗w). The rank-one tensors $v_{S_{r}}⊗v_{T_{r}}⊗v_{C_{r}}$ are the one reaching maximum singular values according to the PTA*k* algorithm used for the multiway correspondence analysis [21], the FCA*k* method.

If the vectors $v_{S_{r}}$, $v_{T_{r}}$, and $v_{C_{r}}$ were non-negative, a simple normalisation would make Equation (Equation 24) a decomposition like a weighted sum of latent joint distributions of independent variables. This is already the case for r=0, as $\left(p_{S}v_{S_{0}}\right)⊗\left(p_{T}v_{T_{0}}\right)⊗\left(p_{C}v_{C_{0}}\right)=\left(p_{S}⊗p_{T}⊗p_{C}\right)$ is the joint distribution of *S*, *T*, *C* as if they were independent and, $H\left(p_{S}⊗p_{T}⊗p_{C}\right))=H\left(S\right)+H\left(T\right)+H\left(C\right)$. For any given r>0, with a non-negative tensor $\left(p_{S}v_{S_{r}}\right)⊗\left(p_{T}v_{T_{r}}\right)⊗\left(p_{C}v_{C_{r}}\right)=\mu_{r}\left(p_{S_{r}^{\prime }}⊗p_{T_{r}^{\prime }}⊗p_{C_{r}^{\prime }}\right)$, with $\mu_{r}=\left(\sum_{s}p_{s}v_{s_{r}}\right)\left(\sum_{t}p_{t}v_{t_{r}}\right)\left(\sum_{c}p_{c}v_{c_{r}}\right)$ and $p_{S_{r}^{\prime }}=p_{S}v_{S_{r}}/\left(\sum_{s}p_{s}v_{s_{r}}\right)$, *idem* for the other components, then $H\left(p_{S_{r}^{\prime }}⊗p_{T_{r}^{\prime }}⊗p_{C_{r}^{\prime }}\right))=H\left(S_{r}^{\prime }\right)+H\left(T_{r}^{\prime }\right)+H\left(C_{r}^{\prime }\right)$. So, the FCA*k* method, after providing the tensor decomposition of the statistic $1+\chi^{2}/N$ with $\chi^{2}$ the weighted distance to 1 of the ratio to independence ($p_{stc}/\left(p_{s}p_{t}p_{c}\right)$), would provide an interpretation of the associations expressed in each optimal rank-one tensors, in terms of additive entropy across the dimensions. Multiway correspondence analysis proposes then an alternative to the mutual information as a metric measuring associations between involved variables. From its set of latent variables, each rescaled rank-one tensor would express a spatio-temporal structuring in interaction with *C* extracted for the initial multiway data table within an independence paradigm. Ratios such as,
(25)$$H\left(S_{r}^{\prime }\right)/H\left(p_{S_{r}^{\prime }}⊗p_{T_{r}^{\prime }}⊗p_{C_{r}^{\prime }}\right)$$
or
(26)$$\left(H\left(T_{r}^{\prime }\right)+H\left(C_{r}^{\prime }\right)\right)/H\left(p_{S_{r}^{\prime }}⊗p_{T_{r}^{\prime }}⊗p_{C_{r}^{\prime }}\right)$$
would highlight the entropic contribution from $S^{\prime }$ to the information structuring extracted from the rank-one tensor.

However, the PTA*k* algorithm used in the FCA*k* method is not a non-negative tensor decomposition, but has the property of providing a nested decomposition (within a hierarchical system) similarly to SVD, which existing non-negative tensor decomposition algorithms (NNTF) do not possess [30]. So besides for r=0, the $v_{S_{r}}$, $v_{T_{r}}$, and $v_{C_{r}}$ will have negative entries, just because of orthogonality constraints set up in the algorithm. However, for each rank-one tensor $\left(p_{S}v_{S_{r}}\right)⊗\left(p_{T}v_{T_{r}}\right)⊗\left(p_{C}v_{C_{r}}\right)$, the tensor:(27)$$CTR_{r}{=}^{def}\left(p_{S}v_{S_{r}}^{2}\right)⊗\left(p_{T}v_{T_{r}}^{2}\right)⊗\left(p_{C}v_{C_{r}}^{2}\right)$$
termed the CTR-tensor, satisfies the positivity and corresponds to a product of distributions as $\sum_{s}{\left(p_{S}v_{S_{r}}^{2}\right)}_{s}=\sum_{s}p_{s}v_{{S_{r}}_{s}}^{2}=1$, from Equation (Equation 23), *idem* for the other components. Each $p_{s}{\left(v_{S_{r}}\right)}_{s}^{2}\;\%$ is a relative contribution (CTR) of the category *s* to the component $v_{S_{r}}$ of the *r*^th^ rank-one tensor, which contributes at $\sigma_{r}^{2}/\left(1+\chi^{2}/N\right)\;\%$ of the whole decomposition or $\sigma_{r}^{2}/\left(\chi^{2}/N\right)\;\%$ to the departure from complete independence used in 2-way correspondence analysis [28] and multiway correspondence analysis (FCAk) [21]. Therefore, $CTR_{r}$ quantifies the role of each combination stc within the rank-one tensor and is expressing its spatio-temporal structuring in interaction with *C*. Ratios such as, $HR_{CTR_{r}}^{\left(S\right)}=H\left(p_{S}v_{S_{r}}^{2}\right)/H\left(CTR_{r}\right)$ or $HR_{CTR_{r}}^{*(S)}=\left(H\left(p_{S}v_{T_{r}}^{2}\right)+H\left(p_{S}v_{C_{r}}^{2}\right)\right)/H\left(CTR_{r}\right)$ highlight the entropic contribution to the relative importance from *S* in the information structuring extracted from the rank-one tensor. Linked the $CTR_{r}$ is the rank-one tensor itself for which a non-negative approximation would allow a similar entropy decomposition.

Instead of using an NNTF, analytic solutions to extract meaningful positive rank-one tensors from an optimal decomposition such as SVD or Equation (Equation 24) have been proposed [31,32], mostly used as initialisation of an NNTF algorithm though with optimality on their own. Following the approach in [31] a rank-one tensor of order k=3 can be decomposed as: $$\begin{array}{ccc} \left(x⊗y⊗z\right) & = & \left(x^{+}-x^{-}\right)⊗\left(y^{+}-y^{-}\right)⊗\left(y^{+}-y^{-}\right) \end{array}$$(28)$$\begin{array}{cc} = & \left(x^{+}⊗y^{+}⊗z^{+}+x^{-}⊗y^{-}⊗z^{+}+x^{-}⊗y^{+}⊗z^{-}+x^{+}⊗y^{-}⊗z^{-}\right)\\ - & \left(x^{+}⊗y^{+}⊗z^{-}+x^{+}⊗y^{-}⊗z^{+}+x^{-}⊗y^{+}⊗z^{+}+x^{-}⊗y^{-}⊗z^{-}\right) \end{array}$$(29)$$\begin{array}{cc} = & {\left(x⊗y⊗z\right)}^{+}-{\left(x⊗y⊗z\right)}^{-} \end{array}$$
where $u^{+}$ and $u^{-}$ are respectively the positive and negative parts of a vector *u*, i.e., $u=u^{+}-u^{-}$ with $u_{i}^{+}=u_{i},\;if\;u_{i}>0$ and =0 otherwise, $u_{i}^{-}=-u_{i},\;if\;u_{i}<0$ and =0 otherwise. From this definition, $\forall i,\;u_{i}^{+}u_{i}^{-}=0$, so $u^{+}⊥u^{-}$. Because of the tensor product and non-overlaps of $u^{+}$ and $u^{-}$, it is easy to see that each non-zero cell in (x⊗y⊗z) comes from exactly one term in the right hand side of Equation (Equation 28), so in one term either in ${\left(x⊗y⊗z\right)}^{+}$ or ${\left(x⊗y⊗z\right)}^{-}$ by then defined. Moreover, as $u^{+}⊥u^{-}$, u∈{x,y,z}, all rank-one tensors involved Equation (Equation 28) are orthogonal by construction. The orthogonality occurs for two vectors of the tensor product in between two rank-one tensors in either ${\left(x⊗y⊗z\right)}^{+}$ or ${\left(x⊗y⊗z\right)}^{-}$, and at least once between rank-one tensors from these two groups. Therefore, ${\left(x⊗y⊗z\right)}^{+}$ and ${\left(x⊗y⊗z\right)}^{-}$ have a minimal non-negative decomposition of maximum r=4 rank-one tensors. For example if $x=x^{+}$, ${\left(x⊗y⊗z\right)}^{+}=\left(x^{+}⊗y^{+}⊗z^{+}+x^{+}⊗y^{-}⊗z^{-}\right)$ and ${\left(x⊗y⊗z\right)}^{-}=\left(x^{+}⊗y^{+}⊗z^{-}+x^{+}⊗y^{-}⊗z^{+}\right)$.

Now, each rank-one tensor $\left(p_{S}v_{S_{r}}⊗p_{T}v_{T_{r}}⊗p_{C}v_{C_{r}}\right)$ in Equation (Equation 24) can be analytically decomposed as ${\left(p_{S}v_{S_{r}}⊗p_{T}v_{T_{r}}⊗p_{C}v_{C_{r}}\right)}^{+}$ and ${\left(p_{S}v_{S_{r}}⊗p_{T}v_{T_{r}}⊗p_{C}v_{C_{r}}\right)}^{-}$ with their respective non-negative rank-one tensors decomposition that can be interpreted similarly to a $\mu_{r}\left(p_{S_{r}^{\prime }}⊗p_{T_{r}^{\prime }}⊗p_{C_{r}^{\prime }}\right)$ above.

## 6. Cartographic Representations of the Quantified Information

Section 2 gave an example of a graphical representation for *C* as expressed by the Shannon decomposition theorem. Within the PsishENT framework, *C* graphical maps but also Si, Sh, Ti and Th graphical maps can be produced at first as categorical maps using the spatial patch and temporal patch background identification, $S_{P}$ and $T_{P}$. For example, a simple coloured geographical map can highlight spatial sizes from Si, for one particular *c* or the $H^{u}\left(Si|C=c\right)$ at each patch with C=c. Considering all *c*’s a map of the $H^{u}\left(C|Si=s_{i}\right)$ at each patch of size $s_{i}$ can be produced, highlighting the heterogeneity in *C*e depending on the patch sizes, or $H^{u}\left(C|Si=s_{i},Ti=t_{i}\right)$ at given specific time size patches. For the latter, it is possible to produce a series of geographical map from reporting $H^{u}\left(C|Si=s_{i},Ti=t_{i}\right)$ at each patch of size $s_{i}$ at each time of a chosen temporal patch of size $t_{i}$. Similarly, at a given patch of $s_{i}$ a time series plot with $H^{u}\left(C|Si=s_{i},Ti=t_{i}\right)$ at each $t_{i}$ can be used.

Various plots can be produced based on background $S_{P}$ and $T_{P}$ references with possible overlaps, with a role similar to the spatio-temporal support of the observations, and then using their categorisations with *C*, Si, Sh, Ti and Th with the entropy decomposition to report the chosen statistics. Endless possibilities of visualisations are foreseen including dynamical plots of $S_{P}$ across time or Ts across space (time series), where time and space may refer to the vision of a ’constant’ support such as *T* and *S* in the Section 3.

The multiway correspondence analysis provides natural ways of plotting spatio-temporal associations across *C* as well as spatial cartographic maps, e.g., spatial or spatio-temporal scores from reconstructing a particular rank-one tensor its CTR.

## 7. The PsishENT Operational Framework

All the previous sections constitute the building blocks of the PsishENT framework which integrate all these aspects within a successive set of choices and analyses. In Figure 1, a generic workflow of using the framework is presented where the three major steps reflect their multiple choices that are detailed in the previous sections.

In **(i)**, after possible transformations of the initial data (not shown here) the definitions of spatial and temporal patches are made, based on rules (i.e., topology, fuzziness etc.), which generate categorical variables Si and/or Sh, Ti and/or Th which may result in classes of sizes or shapes after aggregation rules (Section 4). In **(ii)**, choosing the variables involved (dimensions of the multiway table) and the statistic to compute cell values in the multiway table, includes various choices, i.e., a positive value for each multiway indices, e.g., C=c, Si=1, Ti=3 (Section 3).

The simplest being the number of occurrences, the purpose is to render a multiway distribution like table that is encapsulating the chosen spatio-temporal topological features for *C*. Then, in **(iii)**, a series of analyses based on entropy decomposition theorem (Section 2) and other methods (Section 5) that embed distribution decompositions that are related to for example criteria of independence, homogeneity, uniformity, can be performed to produce results in forms of summary table (e.g., break down of entropy), maps and curves (e.g., time series of a statistic based on an entropy), see Section 6 or from the equations listed in previous sections.

The following section shows various uses of the framework in the context of land cover evolution generated from a climate change simulation. However, the PsishENT framework is adapted to different kind of domains in physical geography, health geography, epidemiology, demography, urban planning or even big data (geolocated social information) for observational or compositional data, as long as concepts of patches, patch sizes and patch shapes would have a meaningful interpretation for the domain. The framework is working with one or more categorical variables observed, measured or simulated spatio-temporally. For quantitative variables, transformations in the first place such as clustering or quantile separation can be applied beforehand.

## 8. Illustrative Example of Land Cover Forecasts

The PsishENT framework offering a range of analyses based on entropy decomposition to highlight spatio-temporal information structuring, the purpose of this example is to show the most simple and illustrative aspects and its flexibility. The data comes from a climate simulation using a Land Surface Model (LSM) predicting the plant functional types (*pfts*) between 2014 and 2100 [33]. Plant functional types describe the vegetation that constitutes the land cover, e.g., boreal broadleaf shrubs, C3 grass. The LSM is driven under a climate forcing scenario, here the RCP8.5 defined by the Intergovernmental Panel on Climate Change (IPCC). RCP8.5 represents a trajectory of concentration of greenhouse gas that would occur for a targetted radiative forcing in 2100, here of 8.5 W/m${}^{2}$; this would mean a global average warming of +3.7 ${}^{\circ }$C in 2050 [34].

For each spatial grid cell (here with a resolution of $2^{\circ }$ of latitude and longitude) a fraction of occupation of each *pft* is estimated within the forecasting at each simulation time step. So, the data used here corresponds to a compositional data. The full list of *pfts* used in the LSM ORCHIDEE (ORganizing Carbon and Hydrology in Dynamic EcosystEms), with the version ORCHIDEE_HLveg [33,35] is given in Appendix. Note that *’bare ground’* is also taken as a *pft*. To come back to an observational data one can transform the data such as considering the dominant *pft* in each of the single grid cell with its fraction as a weight or considering each grid cell as an observation for each *pft* with a weight, i.e., multiple observations for a given *c* (a *pft*), a *pft*, with common spatio-temporal positions. To determine the patches the description using weights was used but dominant categories as summary was also used to represent the data graphically.

Figure 2 displays the distribution (as proportion of cells over a year) of the dominant *pfts*. From the year 2025, the already higher spatial proportion of *pft9* dominance, boreal needleleaf summergreen, than most *pfts*, keeps increasing from 25% to almost 40% in 2099 and in the meantime *pft13*, boreal broadleaf shrubs, decreases from 25% to 10%. From 2039 to 2099, *pft10* dominance, C3 grass, halved, while in the meantime *pft4* doubles and *pft6* increases from 1% to 7%, temperate needleleaf evergreen and temperate broadleaf summergreen respectively. The boreal needleleaf evergreen, *pft7*, shows a sudden drop in 2059 from 5% to 2%, after a drop of 5% between 2014 and 2025 (halved). In Figure 3, the exact evolution of the proportions of occupation for *pft9*, *pft13*, and *pft4* are coherent with what has been described, so far, but the information is not quantified.

Figure 4 confirms spatially the changes observed in Figure 2, from looking at the dominant *pft* per spatial grid cell at the three years 2020, 2050 and 2100. *pft9* is increasing mostly in Russia; *pft13* is disappearing from the Fennoscandia region and southern Russia to appear in northern Russia replacing *pft10* there; *pft4* and *pft6* are replacing *pft10*, *pft7* and *pft13* in the Fennoscandia area.

Spatial patches of size 1, i.e., one grid cell, were created for fractions of a *pft* category greater than 15%. Grid cells belonging to more than one patch (i.e., more than one *pft* category with an occupation greater than 15%) occurred every year with on average a grid cell belonging to 2.2 patches (median is 2, maximum is 6). Then adjacent patches of size 1 for the same *pft* generated spatial patches of various sizes for a given year and a given *pft*. In Figure 5, the temporal evolution of the distribution of patch sizes are displayed where sizes have been grouped into 7 classes: 1, 2, >2, >7, >25, >50, >100, with for example >25 grouping patches of sizes 26, 27, …, 50. From 2050, patches of class size 1 have an important increase with a bump between 2060 and 2080, classes 2, >2 and >25 show a steady increase whilst the number of patches from classes >7 and >50 are relatively decreasing; >100 relatively stable.

The variation in vertical spread at years 2020, 2050 and 2100 in Figure 5 can be linked to the results in Table 1. Indeed in 2020 the curves can be grouped in three: size 1, sizes 2 to >7 and sizes >25 to sizes >100, in 2050 the spread appears less structured and in 2100, size 1 group is important as the grouping sizes >50 and >100. However, much care is needed here as in Table 1 it is the frequencies of grid cells involved in Si and only the number of patches in Figure 5.

For temporal patches the distribution of sizes have a median of 68 a mean of 59 and a third quartile of 87 out of a potential of 87 successive points from 2014–2100 (the total length). *pft1*, bare ground is the *pft* with the most uniform distribution in temporal patches Ti. *Pfts* 4, 10, 12, 14, were represented equally in medium range patch size and high range patch size (very little in small range patch sizes); *Pfts* 6, 7 and 8 were more in medium range patch size than high range patch size (very little in small range patch sizes) whilst *pfts* 5, 9 and 13 were concentrated in high range temporal patch sizes.

In Table 1 is reported at years 2020, 2050 nd 2100 the decomposition of the Shannon entropy using the normalisation relative to a uniform distribution and given in Equation (Equation 8). The closer to 1 $H^{u}$ is, the more uniform the distribution is. Due to the normalisation lines 1 and 2, for example, add up to give line 5, once applied the coefficients e.g., 0.7030593∗(0.4479736)+0.6548033∗(0.5520264)=0.6764207. The spatial patch sizes Si as well as *C* alones show an increase of entropy while Si∣C shows a decrease highlighting the increasing effect of *C* in determining the sizes Si. However, the conditional entropy $H^{u}\left(Si|C\right)$ is already quite low in 2020, highlighting the dependence of Si classes of the sizes of spatial patches from the *pfts* categories in *C*.

From this table (Table 1) and parts involved in the conditional entropies one can represent spatially the heterogeneity due to spatial sizes Si that are revealed by the occurring patch sizes per spatial grid. In Figure 6 are geographically mapped the parts contributing to the conditional entropy for *C* knowing local spatial sizes Si, i.e., the sum of the $p_{Si}H^{u}\left(C|Si=si\right)$ for all the sizes si. The closer to 0 the more homogeneous the distribution of *C* is as due to the spatial sizes involved in the local patches. The theoretical maximum heterogeneity is the value given in Table 1 if all sizes Si were involved, so Figure 6 is mapping the % of that maximum value as indicated in Equation (6). Where there was no patches mapped values are missing and can be interpreted as uniformity in *C* because of no patches found. Changes in homogeneity given the local spatial patch sizes are quite dramatic and shows more changes than only the dominant *pft* recorded per spatial grid as in Figure 4. The two figures are indeed complementary. Over the 2020-2100 period, one obverses in Figure 6 a loss of homogeneity given the patch sizes in the Fennoscandia area whilst a slight increase in homogeneity is seen in western and southern east Russia between 2020 and 2050 followed by a slighter decrease at 2100. Northern Russia shows a decrease in homogeneity between 2020 and 2050 followed by an increase at 2100.

Similarly, in Figure 7 is represented the conditional entropy ratio for $H^{u}\left(Si|C\right)$ where local patches of *C* values were used to map the local effect. Overall over the 2020-2100 period, there was an increase in homogeneity as the conditional entropy is decreasing (see Table 1). Spatially there is an increase in homogeneity of patch sizes given the involved *ptfs* (*C*) in all areas, so either less variation in *pfts* or in their patch sizes.

Integrating time patches sizes Ti can be done in various ways using the PsishENT framework, e.g., decompositions as in Section 3 or using the multiway correspondence analysis (Section 5). The latter one is enabling an additive entropy decomposition of modelled spatio-temporal interaction of *C* from each rank-one tensors. Then, for a chosen time, e.g., 2020, 2050 and 2100, and a chosen class in *C* or the local dominant *C* category (*pft*), a map of a score built at each grid cell from rank-one components weights for *C*, Si and Ti can be used to render the information structuring provided by selected rank-one tensors from the multiway correspondence analysis.The score can be also the corresponding CTR-tensor to render the contributing influence at a grid cell.

Using multivariate occurrences of Si×Ti×C gives a 7×8×11 contingency table analysed by the multiway correspondence analysis. The rank-one tensor of independence of the three variables Si, Ti and *C*, i.e., corresponding to r=0 in Equation (Equation 24) has its components from the multiway table margins, in Table 2 along with other rank-one tensors CTRs also in Table 3.

It represents 40.9% of variability of the data, i.e., $\sigma_{0}^{2}/\sum_{r}\sigma_{r}^{2}$ as expressed in Equation (Equation 24). Large spatial patches, Si>100 and Si>50, are most frequent as well as long time patches, Ti>60, but recording the count of grid cells involved *per* patch size creates an expected monotonic increases. *pft9*, *pft13*, *pft10* and *pft1* are the most frequent patches. Associations across the 3 dimensions (Si, Ti, *C* as *pfts*) are linked to the CTRs and the signs of the coordinates, in the decomposition (Equation 24)) and Equation (Equation 27). Signed CTRs for the rank-one tensors are reported in Table 2 and Table 3. For example, for the rank-one tensor representing 16.7% of variability (or 28.26% within the 60.1% left after complete independence captured by the first rank-one tensor, r=0), Si>100 is mostly associated with *pft13* and *pft9* whilst Si>7 is with *pft12* and to a less extent with *pft5*, all with mostly very long and long time patches, Ti>60 and Ti>30 (the time component is the same as for complete independence). For the rank-one tensor of 9.54% of variability (with the same spatial patch size component), *pft1* is associated with small time size patches 1, 2 and medium sizes >7, opposed to *pft9* with large time patches >60.

For the complete independence, rank-one tensor with 40.9% of variability, which is also the CTR-tensor, the normalised entropy is $H^{u}\left(p_{Si}⊗p_{Ti}⊗p_{C}\right))=1/log(|Si|*[Ti|*|C|])(log(|Si|)*H^{u}\left(p_{Si}\right)+log\left(|Ti|\right)*H^{u}\left(p_{Ti}\right)+log\left(|C|\right)*H^{u}\left(p_{C}\right))=0.764$ with $H^{u}\left(p_{Si}\right)=0.786$, $H^{u}\left(p_{Ti}\right)=0.595$ and $H^{u}\left(p_{C}\right)=0.894$. Therefore, within this 40.9% of variability where large spatio-temporal patches of mostly of *pft9*, *pft13* but also *pft10* or *pft1*, temporal patches are more structuring than spatial and distinction of *pfts*. CTRs entropy decomposition for the first best tensors of the FCA3 optimisation are in Table 4.

Note for the last two tensors (9.54% and 3.55% of variability), the structuring due to *pfts* becomes more important as the entropy for *C* becomes smaller. In Figure 8, maps of the first three CTR-tensors are complementing the quantifications of the information from Table 2 and Table 4. For each grid cell, the geometric mean of the product of the component weights (for the local Si, Ti and *C*) as each score from Equation (Equation 27) were signed with the local *C* component weight in order to highlight the differences in *pfts*. This gives a spatial *intensity* of the patterns of spatial sizes Si, temporal patches Ti and the categorical variable *C* (here the *pfts*). The differentiation due to the sign of *C* weights is useful here but multiple maps per *pfts* could be used instead which would allow not to focus only on the dominant *pft*.

If the Figure 2, Figure 3 and Figure 4 are very informative on the land cover evolution for this data, they do not allow quantification of the different roles of *C* and the spatio-temporal embedding. The PsishENT framework provides this type of information as well help to characterise each influence from other graphical representations. First of all, *pfts* categories have variant patch sizes (time and space). Some *pfts* categories are, along time, increasingly explaining the patch sizes distributions which are related to increased homogeneity e.g., *pft9* (boreal needleleaf summergreen) evolution to larger patches. A tendency to increase of a spatial fragmentation is also quantified (see Table 1) which are localised in Figure 6 and Figure 7. Using correspondence analysis (FCA*k*) or the spatio-temporal multiway table with spatial and time patch sizes with *C* (pfts) enabled a double quantification (Table 2 and Table 4) in specific patterns of associations (each rank-one tensor) and using entropy to evaluate the structuring aspect of the components in the tensors. Spatio-temporal intensity of the effects could be mapped (Figure 8). If fortunately the PsishENT approach allows to retrieve some tendencies seen using simple graphics, the quantification are useful and some hidden patterns can be also detected such as *pft12* (mosses) disappearing the north of Fennoscandia and Russia (see Table 2 and Figure 8).

## 9. Discussion & Conclusions

In order to study the information structuring from a categorical variable *C*, the proposed spatio-temporal entropy framework (PsishENT) makes use of topological characteristics for *C* in time space and geographical space. These characteristics are related to patches sizes, Si and Ti and shape Sh and Th. Then, PsishENT reuses entropy decomposition theorem to derive information quantifications from different choices of multivariate distribution of the characteristics using occurrences, co-occurrences or a local non-negative statistical indice. Complementary to the use of conditional entropy for the decomposition theorem, multiway correspondence analysis fitting the multivariate distribution from a sum of rank-one tensors expresses another alternative decomposition. Quantification of the contribution of each rank-one tensor and its positive approximation allow additive entropy across the characteristics involved in the multivariate distribution. Both quantifications and decomposition of the information can lead to spatio-temporal representations helping to interpret the entropy values.

A land cover evolution data example was used to illustrate some aspects of the PsishENT framework. Examples of quantification of the spatio-temporal information, decomposition and graphical representations were demonstrated, highlighting some principles of the framework. Quantification and sometimes double quantification (from correspondence analysis followed by entropy) can be powerful when comparing different spatio-temporal patterns. Making use of both sizes and shapes would lead to more complex choices that were not looked into for this illustrative example but the framework is generic and flexible enough to adapt to a range of interests and specificities of the data.

The monotonic increase of occurrences due to the number of grid cells involved in classes for larger patches will be even more prominent with co-occurrences or local statistic linked to a proximity assessment. This may be seen as a bias in the framework but induce indeed a topological forcing as the basic natural model of patches. Therefore classes of patch sizes or shapes play here an important role. To cancel off this baseline effect, it is also possible to record in the multiway tables the spatio-temporal patches as basic occurrences, not the cells within the patches. Results for the land cover data example with this other choice and using the same analyses are given in Appendix B where similarities and complementarities with Section 8 are also highlighted. Other scale levels in between the basic units, grid cells for our example and spatial or spatio-temporal patches, e.g., using less stringent rules, could be used, integrating another variable seen as the focused topological support. Multiple scale comparison analysis could therefore be performed using the PsishENT for different scales with the normalised entropy or integrating multiple Si for example.

Looking for the spatio-temporal structuring of more than one categorical variable of interest, say *C* and $C^{\prime }$, is possible using the framework with for each categorical variable a choice of Si and $S^{\prime }i$ etc. and on one hand the entropy decomposition theorem could be applied with various forms, though could become complicated. On the other hand, using the multiway correspondence analysis approach which is indeed linked to the mutual information concept, would provide a double quantification and decomposition in patterns involving both *C* and $C^{\prime }$ which can be evaluated and decomposed using the entropy.

## Figures and Tables

**Figure 1 entropy-21-01112-f001:**
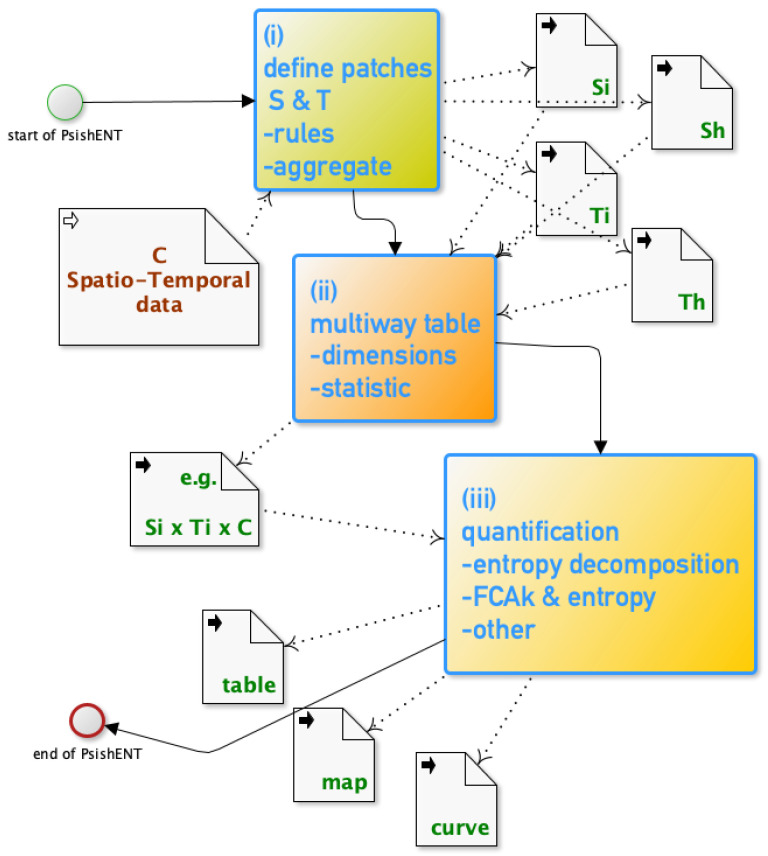
*Modus Operandi* of the patch size and shape entropy (PsishENT) framework.

**Figure 2 entropy-21-01112-f002:**
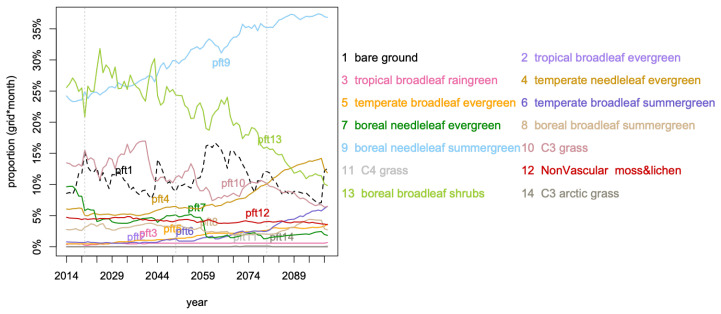
Dominant *pfts* in each spatial grid cell per year.

**Figure 3 entropy-21-01112-f003:**
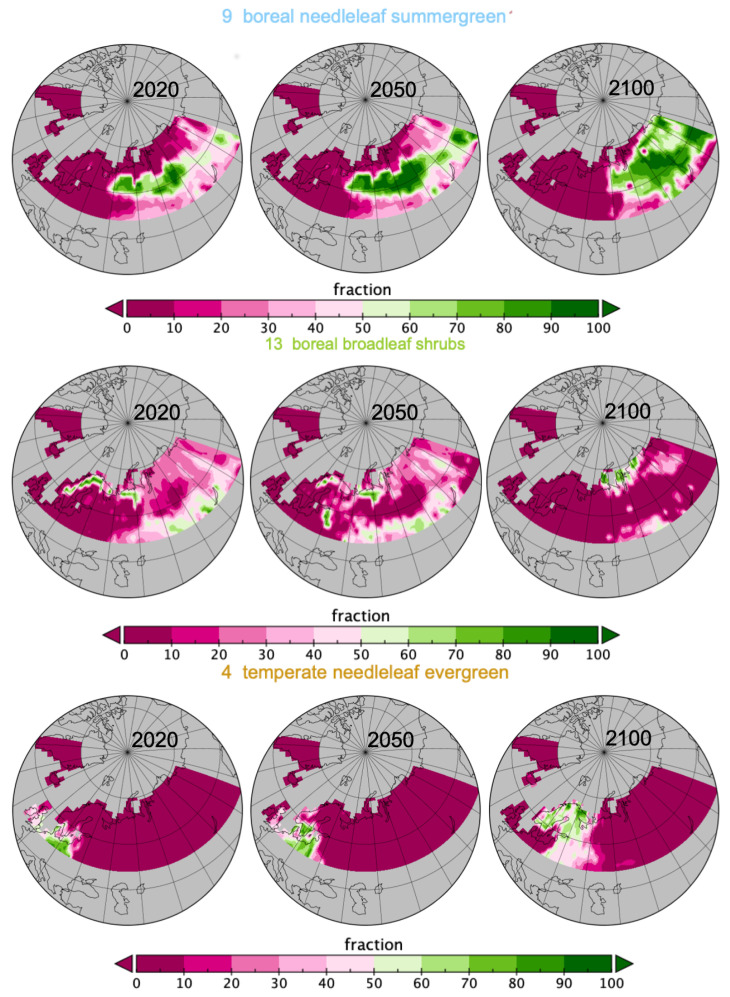
Spatial spread for *pft9*, *pft13* and *pft4* in June for years 2020, 2050 and 2100.

**Figure 4 entropy-21-01112-f004:**
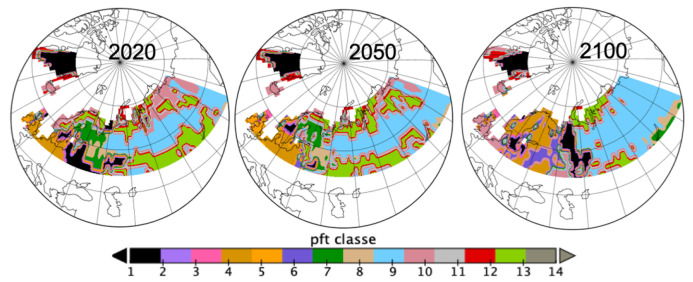
Spatial spread of dominant *pfts* in each grid cell for years 2020, 2050 and 2100 (list of *pfts* given in Figure 2 and in Appendix A).

**Figure 5 entropy-21-01112-f005:**
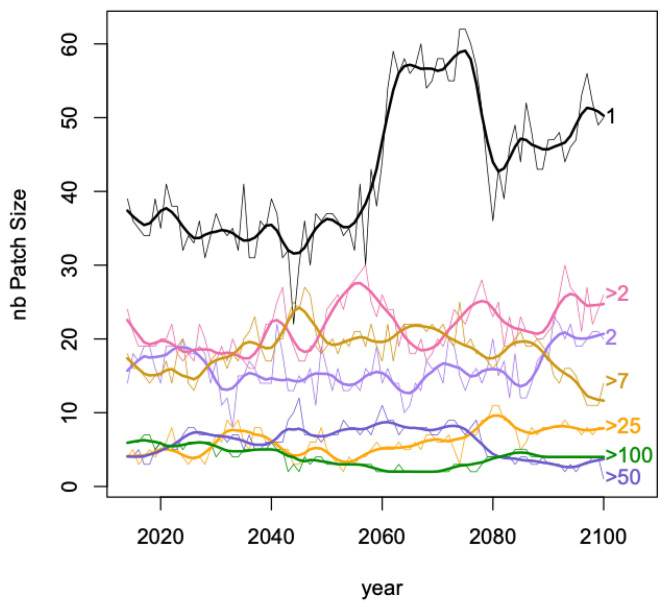
Frequencies of the 7 classes of spatial patches over 846 inland grid cells for all *pfts* where a 1 patch is a grid-cell with fraction >15% (wider solid lines are smoother fit of the time series) in thinner lines.

**Figure 6 entropy-21-01112-f006:**
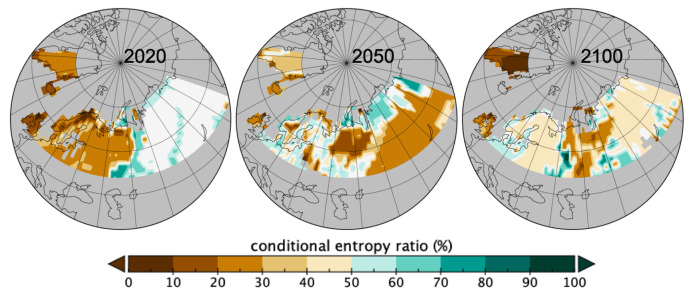
Map of the ratios to conditional entropy $H^{u}\left(C|Si\right)$ of Table 1 from occurring local patch sizes (ranges: 2020 2%–77%, 2050 2%–80%, 2100 0%–92%).

**Figure 7 entropy-21-01112-f007:**
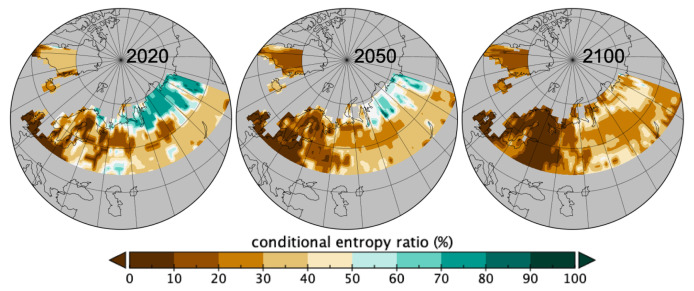
Map of the ratios to conditional entropy $H^{u}\left(Si|C\right)$ of Table 1 from occurring local patches of *C* (ranges: 2020 1%–87%, 2050 0%–89%, 2100 1%–67%).

**Figure 8 entropy-21-01112-f008:**
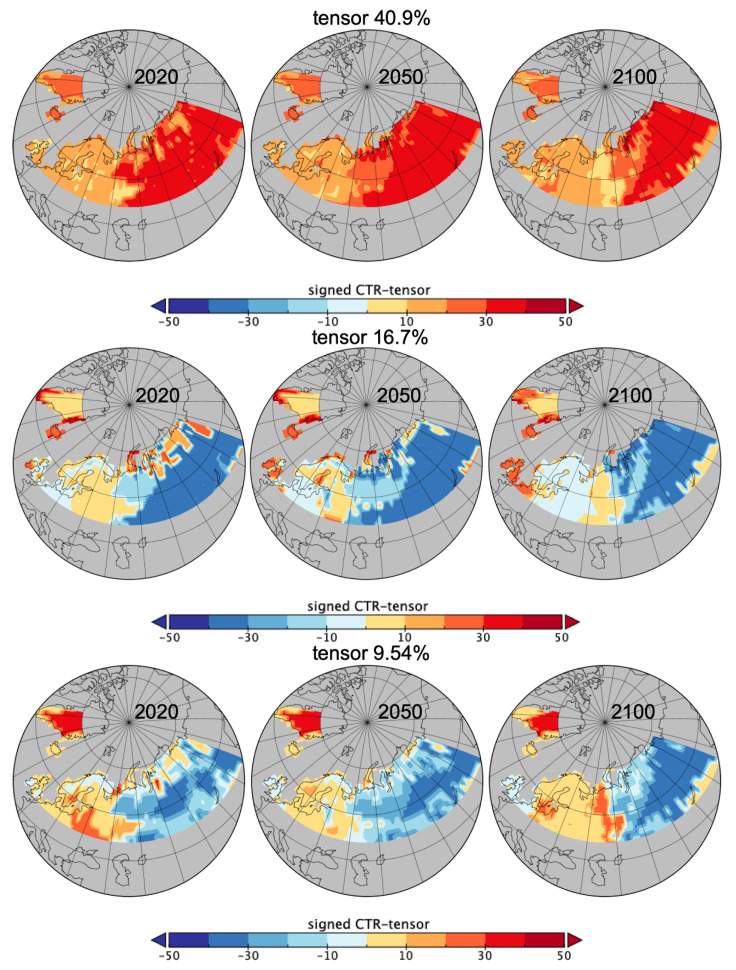
*C*-signed CTR-tensor spatial scores rebuilt for the dominant *pft* in June for years 2020, 2050 and 2100.

**Table 1 entropy-21-01112-t001:** Decomposition of the normalised Shannon entropy, Equation (Equation 8), for the spatial patch sizes classes Si and the *pft* categories variable *C* (11 categories out of 14, see Appendix A) at 2020, 2050 and 2100. ($\frac{log(|S|)}{log(|S|)+log(|C|)}=0.4479736$ and $\frac{log(|C|)}{log(|S|)+log(|C|)}=0.5520264$)

$H^{u}\left(.\right)$	Year 2020	Year 2050	Year 2100
Si	0.7030593	0.7933917	0.7640653
C∣Si	0.6548033	0.5613683	0.6314215
*C*	0.8520292	0.8745148	0.9297879
Si∣C	0.4600228	0.4075092	0.3963961
Si,C	0.6764207	0.6653087	0.6908424

**Table 2 entropy-21-01112-t002:** Margins of the multiway table Si×Ti×C and Signed CTRs (rounded %) for the rank-one tensor of the FC3 representing 40.9% and 16.7% of the variability (see entropy decomposition in Table 4).

Margins & Tensor 40.9%						Tensor 16.7%					
Si		Ti		***C***		Si		Ti		***C***	
1	2	1	1	*pft1*	10	1	3	1	1	*pft1*	5
2	2	2	1	*pft4*	6	2	3	2	1	*pft4*	0
>2	5	>2	1	*pft5*	2	>2	7	>2	1	*pft5*	10
>7	14	>4	2	*pft6*	4	>7	35	>4	2	*pft6*	0
>25	12	>7	8	*pft7*	4	>25	2	>7	8	*pft7*	2
>50	23	>20	7	*pft8*	4	>50	2	>20	7	*pft8*	6
>100	42	>30	21	*pft9*	23	>100	−48	>30	21	*pft9*	−21
		>60	59	*pft10*	13			>60	59	*pft10*	7
				*pft12*	5					*pft12*	21
				*pft13*	22					*pft13*	−22
				*pft14*	7					*pft14*	6

**Table 3 entropy-21-01112-t003:** Signed CTRs (rounded %) for the rank-one tensor of the FC3 of the table Si×Ti×C representing 9.54% and 3.55% of the variability (see entropy decomposition in Table 4).

Tensor 9.54%						Tensor 3.55%					
Si		Ti		***C***		Si		Ti		***C***	
1	2	1	18	*pft1*	60	1	−16	1	27	*pft1*	−81
2	2	2	15	*pft4*	0	2	−9	2	29	*pft4*	1
>2	5	>2	11	*pft5*	−1	>2	−26	>2	14	*pft5*	5
>7	14	>4	9	*pft6*	1	>7	−15	>4	13	*pft6*	3
>25	12	>7	18	*pft7*	3	>25	−5	>7	9	*pft7*	1
>50	23	>20	6	*pft8*	2	>50	7	>20	0	*pft8*	1
>100	42	>30	1	*pft9*	−29	>100	22	>30	−2	*pft9*	0
		>60	−24	*pft10*	0			>60	−8	*pft10*	3
				*pft12*	0					*pft12*	3
				*pft13*	−4					*pft13*	0
				*pft14*	0					*pft14*	2

**Table 4 entropy-21-01112-t004:** CTR-tensor entropy decomposition for the FCA3 of the multiway table Si×Ti×C: the four best rank-one tensors, Equation (Equation 24), representing altogether 70.69% of variability

$H^{u}\left(.\right)$	Tensor 40.9%	Tensor 16.70%	Tensor 9.54%	Tensor 3.55%
Si	0.786	0.661	0.786	0.929
Ti	0.596	0.596	0.908	0.830
*C*	0.894	0.830	0.452	0.356
CTR−tensor	0.765	0.703	0.701	0.683

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
