# Peer review of "On Integrating Size and Shape Distributions into a Spatio-Temporal Information Entropy Framework"

_entropy, 2019, doi:10.3390/e21111112_

Round 1
Reviewer 1 Report
The proposed framework seems interesting but generally, as individual elements are not that novel, the significance seems to be in how the framework is put together, as such it would be beneficial to see some form of diagram/flowchart to illustrate how the framework operates. The examples in section 6 are interesting, but overall it is unclear what added benefit this analysis gives.
The authors have tried to keep to a very general description but at times this makes the paper is hard to follow and unlear, the introduction particularly. It goes straight into technical details with very little context mixing introducing variables which are used later on with introducing concepts. The variables are given as general with some examples of what they relate to but then goes on to talk mainly about ecology. If ecology is where this framework will mainly find application I think it would be easier to only discuss this application.
Also on line 22 should it say "... among n_c (or n_c)"? Should they both be n_c?
Section 2 seems to be largely standard entropy equations applied to the spatial and categorical data, it would be better if the authors introduce the variables S and C in this section.
In section 3 it would be useful if the authors could describe how the overlapping works in practice, they say if the distance to the border needs to "relatively smaller" and that the distance ratio "will not be too affected" but how is this determined is it just from empirical observations?
Similarly, in section 3.1 the authors state that it can be difficult to build a multidimensional distance but given that there are a number of both distance measures and methods for selecting distance measures in higher dimensional spaces it would be interesting to have a comparison of how these impact the performance of the framework.
Author Response
Thank you for your reviews. The replies and actions we have taken to revise the paper are listed in the attached pdf (Reply2Reviewer 1 & 2.pdf).

Reviewer 2 Report
The motivation is not clearly stated in the paper. Why we would like to use the spatio-temporal entropy framework? Is there any parallel method that can be explored for a proper comparison with the proposed framework? I suggest rewriting Abstract and Introduction in a simpler way that reflects the practical need and the implications of the approach. I would expect to have a more storytelling link between the Sections and Sub-sections of the manuscript. At this stage, the Sections are not interlinked and looks like separate Chapters of a book. There are some minor typos that should be fixed.
Author Response

(The authors gave the same response as above.)

Round 2
Reviewer 2 Report
The authors addressed the comments accordingly.